# Climate Change, Agriculture, and Biodiversity: How Does Shifting Agriculture Affect Habitat Availability?

Mary Ann Cunningham 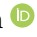

Department of Earth Science and Geography, Vassar College, Poughkeepsie, NY 12604, USA; macunningham@vassar.edu

**Abstract:** Models show that climate change is likely to push agricultural production in the US region known as the Corn Belt northward in coming decades. The economic and social impacts of this northward shift have received extensive attention, but its environmental impacts, such as effects on biodiversity, have received less focus. The aim of this study was to evaluate the extent and distribution of grassland-type habitat that is vulnerable to a northward-shifting Corn Belt. To analyze this question, geographic shifts in suitable climate conditions for the dominant crop, corn (*Zea mays*), were modelled. The amount and distribution of uncultivated (potential habitat) land cover classes was then calculated and mapped in current and future (2050) regions suitable for corn. In currently-suitable areas, the degree of climate suitability positively predicted the dominance of corn in the landscape and negatively predicted grasslands. Areas likely to become climatically suited for corn production contained modest amounts of grassland and herbaceous wetland, most of it privately held and lacking protected status. If economic incentives for corn remain strong, pressure to further simplify the landscape and further reduce habitat resources will likely increase in the coming decades. While global concern for biodiversity and habitat conservation is growing, this study raises the question of how wealthy countries are taking action, or not, to reduce further land conversion and habitat losses.

**Keywords:** climate change; Corn Belt; cropland data layer; grassland biodiversity; Maxent





## 1. Introduction

Climate change poses important risks to agricultural regions, among them, the central U.S. region known as the Corn Belt, which has substantial influence on national and global economies and food systems [1]. Corn (*Zea mays*) dominates agricultural production in this region (over 250 million metric tons/year), followed by soybeans (*Glycene max*; over 70 million metric tons/year) [2]. In this agriculture-dependent region, climate change is expected to have important economic and social impacts. These impacts are associated with economic stability, rather than food security, as most US corn crop is used for the production of ethanol (for transportation) and as feed for beef, pork, and chicken production. The Corn Belt, also called the Corn–Soy Belt, is among the most productive farming regions in the world. Its high production depends on a base of organic-rich soils, generally high summer precipitation, advanced production technology [3], and a climate well suited to corn production in most years. Suitable climate conditions have been an important advantage, as states that lead corn and soybean production are able to produce 90 percent of corn and 95 percent of soybeans without irrigation [2] (Figure 1).

Climate conditions strongly affect the economically important corn crop, as corn is sensitive to high temperatures [4–6]. Consequently, numerous studies have modelled likely changes in corn distribution and yields in this region. In general, climate change is expected to shift the region of concentrated corn production northward, as summer temperatures rise [7,8]. Yields and production quantities are likely to decline in higher temperatures [7,9,10]. Farmers in the region are expected to face income losses [11], although

Arbuckle et al. [12,13] and Leiserowitz et al. [14] have found that farmers themselves often do not consider climate risk a personal concern. Despite climate scientists' projections of declining and shifting production, ethanol producers, which consume over half of the US corn crop [3], express confidence in plans for long-term expansion [15,16].

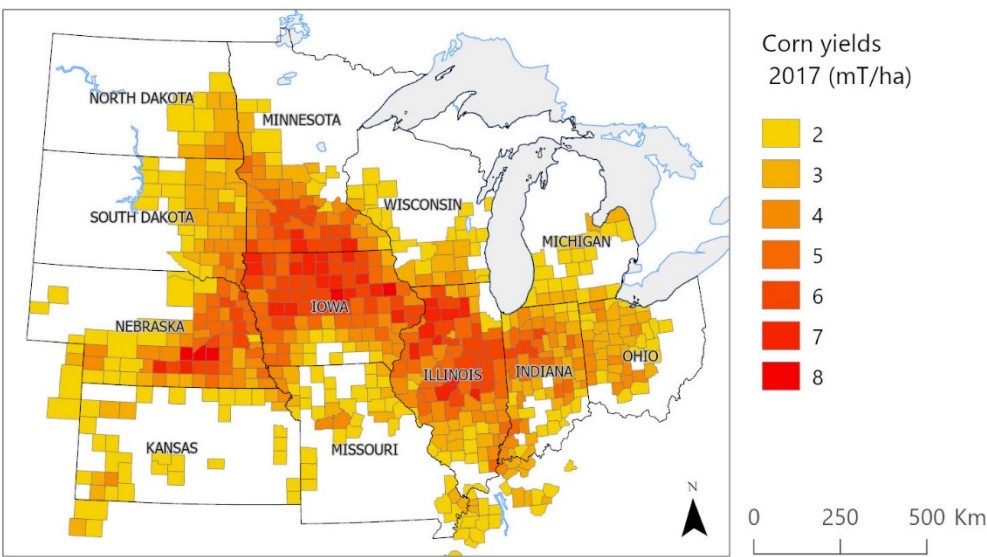

**Figure 1.** Leading Corn Belt states, with yields from 2017 census of agriculture.

While the human impacts of geographic changes in corn production have received close attention, fewer studies have examined the impacts of a northward-shifting Corn Belt on non-human ecosystem values and services, such as biological diversity. The biodiversity in the remnant grasslands, shrublands, and herbaceous wetlands in the region, in particular, provides critical breeding habitat for migratory grassland and wetland birds, as well as supporting invertebrate and floral diversity [17–20]. Despite over a century of cultivation, the upper Midwest supports much of the Western Hemisphere's migratory waterfowl population, as well as prairie songbirds and upland gamebirds that persist in fragments of remnant grassland, herbaceous wetland, and shrubland habitat [20–22].

A continental-scale shift in this region's agricultural production has global implications, both for ecosystem services and for biodiversity. Governments have agreed that protecting habitat areas is key to saving ecosystems and reducing global warming, establishing aims to protect 30 percent of the world's surface (often called "30 by 30" goals) and to ensure sustainable management on another 20 percent [23,24]. Much of the focus on conservation has been on low income and developing regions, which still retain extensive uncultivated and un-urbanized lands. In these areas, agricultural expansion has been shown to reduce biodiversity, for example, with expanding oil palm cultivation [25] or food production [26–28]. While these impacts have long been important concerns for conservation, agricultural expansion in wealthier countries has received less academic attention. Disregarding the role of wealthier regions in global biodiversity goals can be seen as shifting responsibility to low-income and developing regions, an obvious injustice [29]. Attention to shifts in production of a major crop such as U.S. corn, then, is important both for regional ecosystem values and for global biodiversity and climate goals and strategies.

Grassland biodiversity in the Corn Belt and surrounding regions persists largely in habitat provided by uncultivated areas of grassland, hay, and herbaceous wetlands. Set-aside farmlands in programs such as the Conservation Reserve Program (CRP) also provide important habitat [22,30,31]. These remnant pieces of uncultivated land cover can support biodiversity [21,32], even when small and fragmented [33,34]. Conservation programs, however, have historically competed poorly against the economic weight of corn production. For example, programs such as CRP, introduced to protect soils and surface waters (and secondarily biodiversity) declined by over 20 percent after the federal

ethanol subsidies in 2005 and 2007 led to rising corn prices [35,36]. In an assessment of land cover data in the USDA Cropland Data Layer (CDL), Alemu et al. found that uncultivated habitat landcover has declined in the region, while field crops, chiefly soy, alfalfa, and corn, increased in recent years in response to incentives for production of ethanol, dairy, and meat livestock [37].

Given that the agriculture industry expects to expand corn production and that climate models show production shifting northward, what will be the impacts on "receiving" areas of an expanding Corn Belt? In particular, how much uncultivated land cover (i.e., potential wildlife habitat) is at risk of being lost to expanding corn production? While it is not possible to know how land cover will change in future decades, we can calculate the extent of uncultivated land uses in areas projected to become highly suitable for corn production. In this study, then, we addressed this concern by (1) modelling the geographic extent of area that is climatically suitable to corn production, both currently and in 2050, and (2) calculating the amount and geographic distribution of uncultivated land cover classes in those two time periods.

## 2. Materials and Methods

To evaluate the extent of uncultivated land uses in future climate-suitable areas for corn production, we proceeded in two stages. We first used climate variables occurring in high-yielding areas to model the geographic range of climate suitability for corn in the recent past and in 2050. Within those geographic ranges (current and 2050), we then assessed the distribution of uncultivated land cover classes.

### 2.1. Study Area

Analysis focused on areas with strong climatic suitability for corn production. Unirrigated yields in these counties are high because of generally reliable growing-season rainfall, rich organic soils, and level glacial topography, among other factors. Much of the region historically had abundant shallow wetlands, and the region spans a gradient of annual moisture from prairie grasslands to eastern deciduous forests, with precipitation ranging from approximately 500 to over 1200 mm annually. Most grasslands and wetlands in the region, and historically much of the forested land cover, have been converted to croplands, and the majority of land use is agricultural [1].

Other factors beyond climate, such as soil conditions, topography, or water bodies, influence the distribution of crop production, but in this region, climate is a key driver [1,3,10]. Wet soils and shallow wetlands have largely been managed or eliminated by field drainage. Inputs, especially fertilizer, adjust for variations in soil quality. Many aspects of environmental variation, then, have been controlled in this environment. Climate conditions can also be accommodated, in that both drought and extreme heat can be mitigated by irrigation. However, a key historical advantage to production in this region has been the ability to produce high yields without irrigation. Climate conditions have thus been an economic advantage to production.

### 2.2. Climate Suitability Modeling

Following previous studies modelling changes in the extent of suitable climate conditions, we used a species distribution model (SDM) to create a model of the current extent of suitable climate conditions from observed high-production counties. An SDM uses environmental variables, together with observed of occurrence locations for a species or crop, to map the geographic range of suitability for those environmental variables. Occurrence locations are treated as environments where the species of interest can occur, and other areas with similar combinations of environmental variables are then identified. While a variety of SDM approaches exist, the program Maxent has proven to be a robust way to describe and predict ranges for both wild species [38], and cultivated crops [39–42]. A machine learning algorithm, this approach seeks the maximum entropy, or best probability

distribution based on available knowledge of the system (i.e., environmental variables and presence observations) [38].

We used the SDM model Maxent [38,43], which produces output suitability values range from 0 (unsuitable) to 1 (highly suitable). For observed occurrence locations, we used a set of 2000 points randomly generated, with a minimum distance of 10 km apart, from within all US counties with less than 10 percent irrigation and with greater than median corn yields (tons/ha) for any of the last 5 censuses taken by the USDA Census of Agriculture [44]. These censuses are conducted in five-year intervals, so these counties represented high-yield, low-irrigation years from 1997 to 2017. This approach produced an inclusive set of counties, because yields may vary in an individual census year.

For input environmental variables used to generate a model of current "suitable" climates, we used WorldClim, current climate (1950−2000), 5 arcmin (~10 km) [43,45]. Nineteen bioclimatic variables [46] were used as input environmental variables. We evaluated correlations among explanatory factors to ensure that leading factors were not strongly correlated, although Elith et al. have argued that multicollinearity is less of a problem for machine learning approaches such as Maxent than it is for standard statistical approaches [47,48].

The "current" Maxent model was then applied to future climate conditions representing 2050, using the representative concentration pathway of 8.5 W/m$^2$ of radiative forcing (RCP8.5) [49]. While there is debate about the most likely trajectory for climate change, Schwalm et al. (2020) have argued that this model corresponds to likely rates of accumulation of greenhouse gases in the atmosphere [50]. For future climate models, we used the same bioclimatic variables for 5 different climate models, then averaged the output to represent an aggregated view of climate conditions for 2050. The five models used and their sources were as follows: NASA Goddard Institute for Space Studies (GISS-E2-R), U.K. Met Office Hadley Centre Global Environmental Model (HadGEM2-ES), the Japan Agency for Marine-Earth Science and Technology (MIROC 5), Max Planck Institute for Meteorology (MPI-ESM-LR), and the Meteorological Research Institute (MRI-CGCM3) [51].

Test statistics generally used to evaluate Maxent output include area under the receiver-operating characteristics curve (AUC) and true skills statistic (TSS; [47,52]). The AUC reports the relative rate of model sensitivity (the proportion of correctly predicted presences) and specificity (the proportion of correctly predicted random pseudoabsences). The TSS is calculated as (sensitivity + specificity − 1), as discussed by Allouche et al. [53].

*2.3. Landcover Assessment*

To evaluate land cover in current and future corn-suitable regions, we delineated areas with suitability > 0.5 for both current and 2050 model output. Within those current and 2050 ranges, we then used Cropland Data Layer (CDL) data [54] as input land cover data in order to calculate the amount of cultivated and non-cultivated land cover. The occurrence of uncultivated classes helps to indicate persistence of uncultivated classes in areas currently suitable for corn production. Assessing the extent of uncultivated classes for 2050 indicates the extent of land cover that could be transformed to cultivation, as climates become more suitable for corn cultivation.

The primary concern of this project was land cover that could contribute to grassland-type habitat, so land cover assessment focused on four northwestern states that contained both the region of future climate-suitability and historically abundant amounts of grassland land cover. Therefore, land cover evaluations focused on four northwestern states in the region: Iowa, Minnesota, North Dakota, and South Dakota (Figure 1).

The CDL has high spatial and attribute detail, with a 30 m cell size and over 50 crop classes, and it is consistently available across the study area [37,54]. To calculate changes in the amount and distribution of uncultivated land cover types, we aggregated CDL classes for the year 2020 into four cultivated classes (corn, soybeans, wheat, other crops) three grassland-shrubland classes (grass/pasture, hay, shrublands, herbaceous wetlands), and three uncultivated non-grassland classes (forest, woody wetlands, and open water). We

then used the Tabulate Area function in ArcGIS Pro 2.9 [55] to sum the area of each land cover class in current and 2050 "suitable" areas.

Protected lands are important in maintaining habitat areas across the region, so protected areas and unprotected areas were evaluated separately. Multiple kinds of conservation programs exist in the region, including state and federal lands, conservation easements on private lands, private conservation areas, wildlife management areas, and designations that protect habitat areas from conversion to croplands. To assess the proportion of land cover classes with any class of protected status, we used protected areas data from the USGS Gap Analysis Program at the federal and state levels [56].

### 2.4. Distribution of Land Cover Classes

In addition to total abundance of uncultivated land cover classes in current and 2050 climate-suitable areas, it is important to visualize and assess the geographic distribution of different land cover classes across the regions. Visualizing distribution of CDL land cover classes across multiple states is difficult, especially for minority classes such as grasslands or wetlands, because the CDL is highly detailed, with 30 m cell resolution. Therefore, to visualize distribution of land cover classes, we aggregated CDL landcover area using the Tabulate Area function to sum all land cover classes in each selected county. This allowed mapping the percentage of each land cover class by county. We also calculated the average suitability value by county in order to plot the percentage of different land cover classes against mean suitability for counties.

## 3. Results

### 3.1. Model Output and Suitability Ranges

The most influential Maxent model variables were mainly temperature variables, reflecting the importance of rising temperatures as well as the small variation in precipitation across this region. The six variables with the greatest relative contribution in the model were mean temperature of the warmest quarter (BIO10, 22 percent contribution); isothermality, or diurnal temperature range/annual temperature range (BIO3, 16 percent); mean diurnal temperature range (BIO2, 15 percent); maximum temperature of the warmest month (BIO5, 9.5 percent); precipitation of the warmest quarter (BIO18, 9 percent); and minimum temperature of the coldest month (BIO6, 7 percent). These six variables accounted for 79 percent of the variation in the model. Variables had correlations of $r < 0.5$, except BIO10 and BIO3, which were correlated with $r = 0.75$. Model explanation was strong, with an AUC of 0.96 and a TSS of 0.85.

The region of future climate suitability was smaller than the current suitable region, and it was farther north and west. The future-suitable region did not include most of current Corn Belt states (Figure 2). Within these regions, the area with suitability > 0.5 occurred in central and northern Minnesota and eastern North Dakota, as well as in the Canadian provinces of Ontario and Manitoba. Low to moderate climate suitability persisted in the northern reaches of the current Corn Belt, however. Currently forested areas of northern states (Minnesota, Wisconsin, Michigan) were projected to become similar, climatically, to conditions currently in southern Iowa or Illinois.

### 3.2. Land Cover

In currently suitable areas (where suitability > 0.5), two-thirds of the landscape was cultivated (excluding "developed" land cover classes; Figure 3). Corn (32%) and soybeans (29%) dominated cultivated areas, followed by wheat (2 percent) and other crops (3 percent). Among uncultivated land cover classes, grassland and pasture were dominant (15 percent), followed by forest (6 percent) and herbaceous wetlands (5 percent). Hay made up 1 percent, and shrublands were negligible.

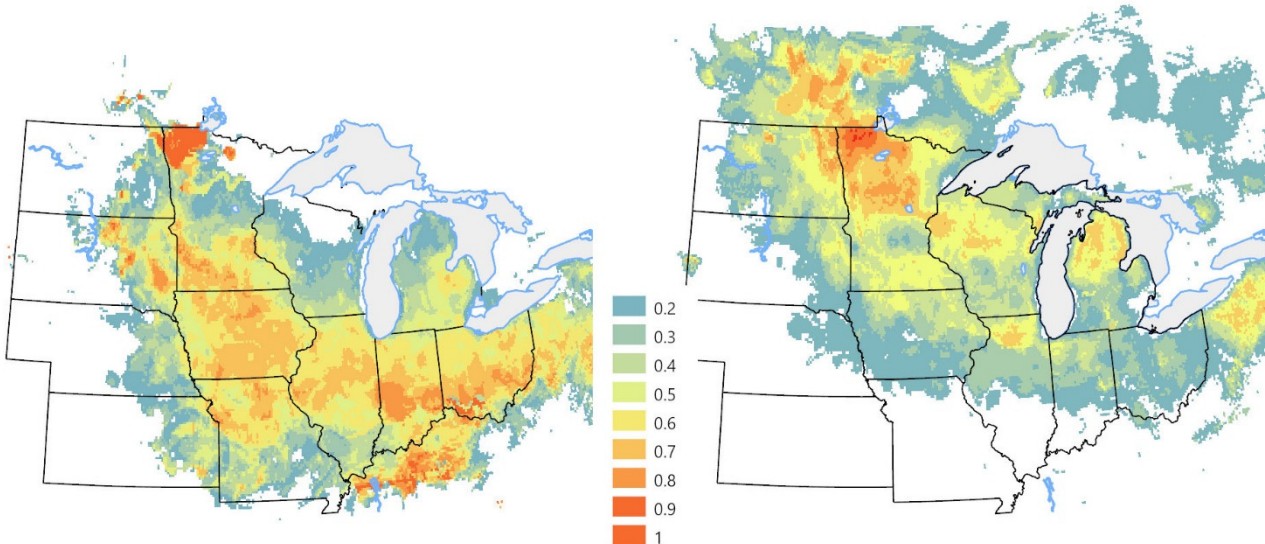

**Figure 2.** Climate suitability model for recent (2050–2000) conditions, left, and 2050, right. Suitability values, shown in center scale, range from 0 (unsuitable) to 1 (highly suitable).

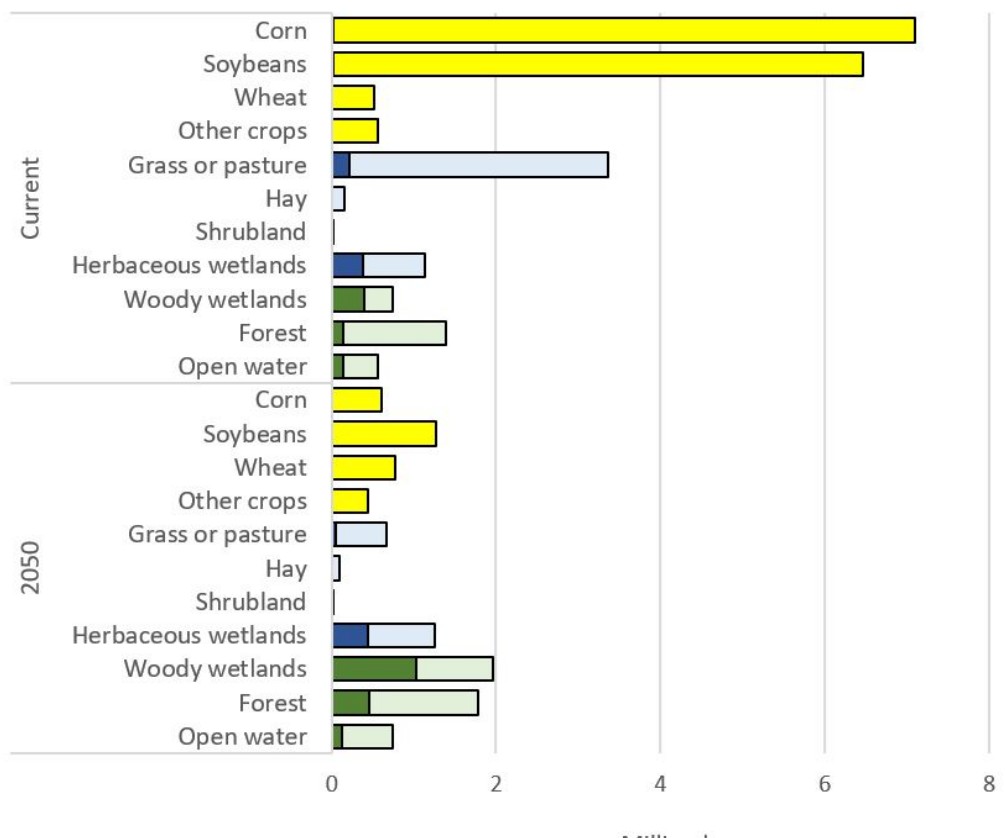

**Figure 3.** Total amount of land cover classes in recent and future areas with suitability > 0.5. Areas with conservation protection status are shown in dark blue (grassland-type land cover) or dark green (forest and water land cover).

In future-suitable areas, one-third of the landscape was in cultivation (32 percent). Corn made up only 6 percent of land cover, with soybeans (13 percent), wheat (8 percent), and other crops (5 percent) comprising the rest. Grass, pasture, and hay together made up 21 percent of the landscape. Woody wetlands (20 percent), forest (19 percent), and open water (8 percent) made up the rest of the landscape.

Proportions of protected areas were similar for all land cover classes in currently-suitable and future-suitable areas (Table 1). The land cover class with the greatest levels of protected status were woody wetlands (53 percent protected) and herbaceous wetlands (33 percent). Grassland habitat classes had little protection in both current and future areas. Only 6 percent of grass or pasture had protected status in currently suitable areas and only 7 percent in 2050. Shrubland had a higher percentage in protected status in both years, but the total extent of shrublands was small across the region. Herbaceous wetlands were the best protected "grassland" habitat, with 33 percent (currently) or 35 percent (2050) protected.

**Table 1.** Percentages of land cover classes with protected status (or not protected) in areas where suitability > 0.5 either currently or in 2050.

| Land Cover | Current | | 2050 | |
| --- | --- | --- | --- | --- |
| | Protected | Not | Protected | Not |
| All | 6 | 94 | 22 | 78 |
| Shrubland | 17 | 83 | 20 | 80 |
| Hay | 2 | 98 | 2 | 98 |
| Grass or pasture | 6 | 94 | 7 | 93 |
| Herbaceous wetlands | 33 | 67 | 35 | 65 |
| Woody wetlands | 53 | 47 | 52 | 48 |
| Open water | 26 | 74 | 17 | 83 |
| Forest | 10 | 90 | 25 | 75 |
| Wheat | 0 | 100 | 0 | 100 |
| Other crops | 1 | 99 | 1 | 99 |
| Soybeans | 0 | 100 | 1 | 99 |
| Corn | 0 | 100 | 0 | 100 |

*3.3. Distribution of Land Cover Classes*

There was a strong correlation between mean suitability value and corn production (Figure 4). In contrast, there were negative correlations between suitability values and percentage of grass or hay in the landscape. Where climate suitability was high (>0.5), the percentage of corn in the landscape reached 40–50 percent in many cases (Figure 4, top graph). Most of the hay/grass classes in currently suitable areas is in South Dakota and North Dakota (Figure 4, lower two maps). In these areas, where climate suitability for corn was especially poor, grass and pasture reached 80 percent of the landscape (Figure 4, middle graph). Hay production was modest in all areas, however, reaching 10 percent in only one of the counties analyzed (Figure 4, bottom graph). In areas most suited to corn production, Iowa and southern Minnesota, abundance of hay/grass land cover classes was small.

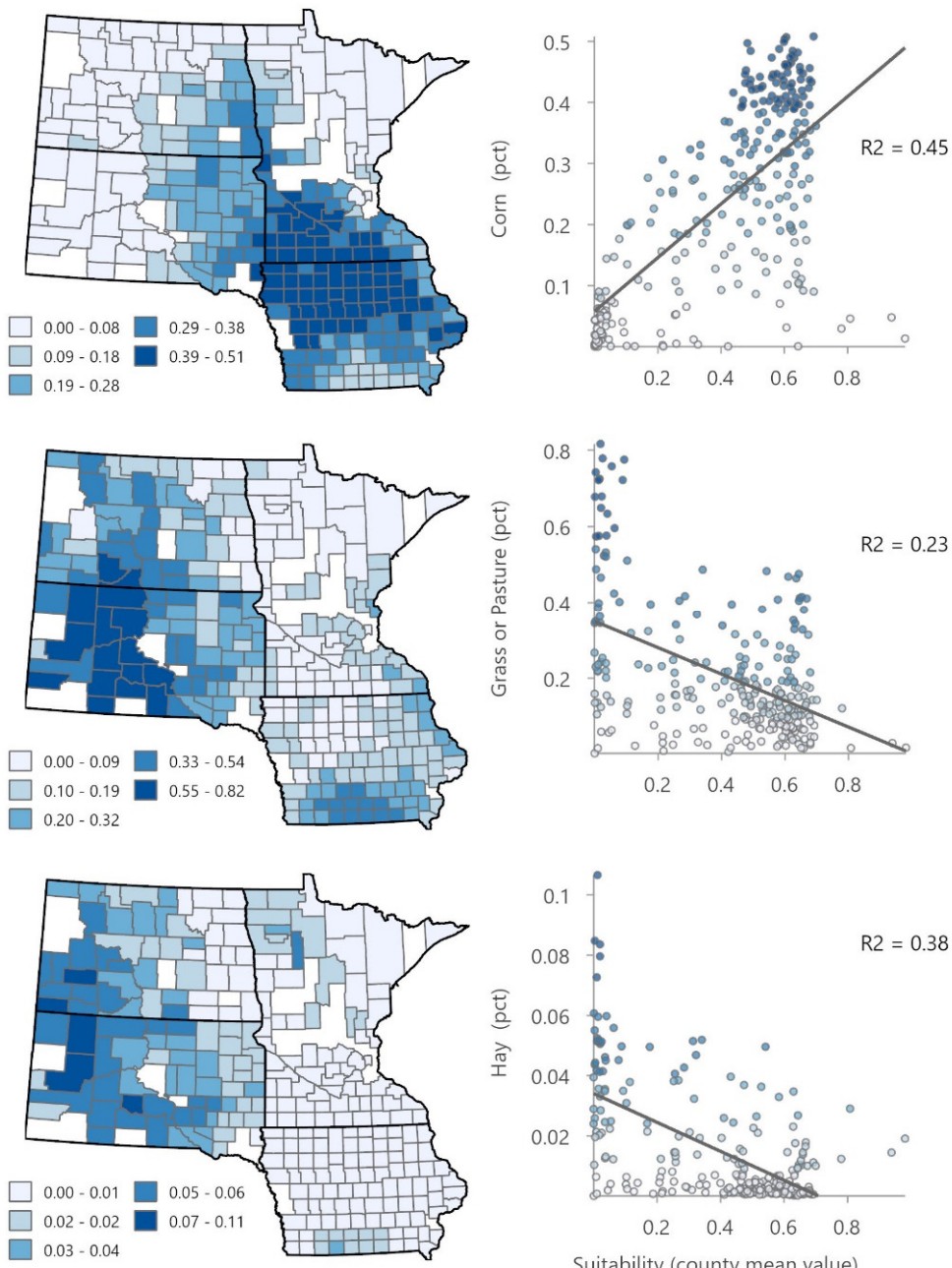

**Figure 4.** Distribution of corn (**top**), grassland + pasture (**center**), and hay (**bottom**), aggregated by county (**left**), and plotted against county mean suitability value (**right**). Values are given as a fraction, where total land cover = 1.0.

## 4. Discussion

### 4.1. Shifting Agricultural Geographies

The motivating question in this study was whether an expanding Corn Belt threatens uncultivated lands that could serve as wildlife habitat. In this study area, grassland habitat is limited but is more abundant than in areas highly suited to corn production. Where climate is strongly suitable for corn, it tends to be dominant, so a northward-shifting Corn Belt is likely to displace uncultivated classes such as hay and grassland, as well as other cultivated classes. Much of the currently existing grassland and pasture land cover is in western reaches of the study area. These areas are too dry for most corn production, and they are unlikely to become more suitable for unirrigated corn production. However, as

the climate warms, if irrigation resources are available, corn expansion could occur in these areas. Already, corn production has expanded in North and South Dakota [2].

The northward shift in the Corn Belt modelled here corresponds to shifts found in previous studies [7,8,11]. While different modelling approaches use different input data and algorithms, there is a strong similarity in the direction and the geography of expected future conditions. There is good agreement, then, on the likely trend of production shifts for this crop region. Moreover, the model used here was a good fit in that it provided strong explanation (AUC = 0.96 and TSS = 0.85). Because climate factors strongly predict current and historical corn production, it seems reasonable to anticipate that climate conditions will remain important in the future. Even in a wealthy country, where crop systems are highly developed, there is vulnerability to climate warming. A cultivated landscape is clearly a complicated system, with economic and social drivers in addition to environmental drivers, and technological fixes may allow this production system to persist in southern regions [7,13], but there is good agreement that northward expansion similar to that found here is likely.

The landscape here is already largely converted to anthropogenic uses. In the present study area, corn and soybeans may displace other crops before they displace remnant pasture, grassland, or hay land cover. However, pasture is less lucrative, in current agricultural systems, than a subsidized product such as corn or soybeans. Pasture and grassland would appear, from past patterns, unlikely to persist as the climate becomes more suitable for major crops such as corn and soybeans.

Economic pressures have historically contributed to declines in voluntary conservation programs such as the Conservation Reserve Program [35–37], and already uncultivated lands are few across the region discussed here. The fact that most grassland-type land cover has no protected status, and that pasture is often uncompetitive if cropping is possible, suggests that these habitat areas are likely to be at risk in future decades.

*4.2. Biodiversity Conservation*

In a context of global concerns about biodiversity and landscape-scale habitat protection, as represented by international "30 by 30" goals, in particular, there is considerable attention to climate change impacts on biodiversity. At the same time, there is attention to vulnerabilities in agricultural production. Bringing these two streams of evidence together is important for perceiving the whole picture of habitat and biodiversity conservation in a changing environment. The follow-on effects of agricultural changes are a central challenge in biodiversity conservation.

Grassland habitat in this region is important but limited in extent, and little of it has protection from further conversion to cultivation. Where climate is strongly suitable, corn tends to be dominant (Figure 4), so a shifting corn belt will likely displace other crops, but it is also likely to displace other "uncultivated" classes such as hay and grassland, as long as incentive structures for corn and soybean production remain as they are. Currently, there is no difference in proportion of protected status in the current- and future-suitable regions. Moreover, despite variations in soil types or topography over the region, climate suitability has a dominant influence on cultivation patterns. Uncultivated land cover classes, then, have limited protection, in the face of increasing economic pressures, as the climate becomes increasingly suitable to corn production. If future decades, grassland habitat may also expand northward, into regions that are currently too cool or moist for this ecosystem, but pressures for agricultural production, already high in the Canadian Prairie Provinces [18], are likely to shift as well.

This analysis did not address forest cover and wetlands, but impacts on these and other habitat types are also uncertain as climate warming transforms forest ecosystems, or if crop prices and cropping practice incentivize the clearing of woodlots or other land conversion. Most wetlands are legally protected in the United States, but if wetlands dry, their protected status may become more tenuous [19]. Moreover, policies to protect

wetlands can and have changed historically. Wetlands, especially drying wetlands, then, may also be vulnerable to shifting economic uses of the landscape.

This study attempts to highlight the impacts of changing agricultural geographies on environmental values. Biodiversity conservation is one of these values, but similar questions arise for water resources, water quality, pollinators, soil conservation, and more [17,18]. As greater confidence emerges on the shape of future agricultural production regions, attention to myriad ecosystem services and environmental values deserves attention.

### 4.3. Incentive Structures and Agricultural Production

Agricultural production patterns in the United States are largely driven by policy structures that incentivize one crop over another [36,37,57,58]. For example, corn production has increased since Farm Bill ethanol supports in 2005 and 2007 encouraged increased ethanol production, as well as policies that facilitate large-scale meat production. More recently, funding for carbon capture and sequestration projects has provided another income stream for ethanol in the form of carbon capture [59].

At present, 90 percent of Iowa corn goes to these two industrial production systems: in 2021, 57 percent of Iowa corn was used to produce nearly 27 percent of US ethanol, and 33 percent went directly into livestock feed [57]. The concern in this system is not efficient production of food: one pound of beef requires at least seven pounds of corn (plus inputs); one pound of pork requires at least four pounds of corn [57]. Notably, Rasche et al. [27] have shown that reducing meat production could increase food supplies while dramatically lowering the environmental impacts of crop production in a changing climate. Food sources other than livestock produce both protein and calories with far lower land costs and other environmental costs [28]. In the ethanol industry, cost–benefit studies of energy, water, and carbon balances have long questioned the ethanol industry's net environmental impacts and ability to persist without subsidies [58].

In terms of agricultural policy, then, it is important to disambiguate food security from industrial production. The Corn Belt is an industrial landscape that produces industrial inputs for transportation and large-scale meat products. As such, it does not directly address global concerns about providing healthy and sustainable food systems [27,28].

Policy structures have created a corn production system on which many rural regions now depend and which differs considerably from historically diversified agricultural production. Rural economies have been tied to a few major crops for decades, but additional options exist. In addition to more diversified food systems, renewable energy is emerging as a stabilizing economic opportunity, for example. Wind power provides an important stream of rural income, and solar power is providing critical income in many farming regions. Emerging practices of agrivoltaics, combined photovoltaic and agricultural production, for example, can stabilize rural income while reducing soil erosion and protecting water quality. Reducing subsidies for biofuel production would lower the pressure to expand corn production, as would policy structures to support a more diversified production system than the current corn–soybean production model. Conservationists concerned about protecting wildlife and its habitat in agricultural regions, then, could pursue a wide range of policies to support diversification, reduce biofuel development, and change the direction of the Corn Belt's development.

### 5. Conclusions

Climate change is likely to push the region of climatic suitability for corn production northward in the coming decades. In the region with current high suitability for corn production, corn and soybeans dominate the majority of land use, and their dominance in the landscape correlates with modelled climate suitability across the region. If economic incentives for these two crops remain strong, they are likely to encroach on areas northward of the current Corn Belt. In these areas, uncultivated land cover persists, although its abundance is low, and continues to support biodiversity. As with corn, the extent of these land cover classes varies inversely to the suitability of climate conditions for corn production.

Attention to competing needs for uncultivated lands will be important as climate change leads to changing geographies of production in the coming decades. Most native habitat has already been displaced by other crops, and an expansion of corn will likely displace those other crops, as the economic value and viability of corn expand northward. That said, uncultivated lands, grass, and pasture do persist, especially where climate conditions are poorly suited to corn and soy production. More study of these follow-on effects of climate-induced crop shifts is needed, as well as attention to policies and incentives that could influence the trajectory of the region's land use and biodiversity in future decades.

**Funding:** Collins Class of '42 Environmental Science Research Fund, Vassar College: 2021.

**Data Availability Statement:** All data are available from sources cited; no new data were created in this study.

**Conflicts of Interest:** The author declares no conflict of interest.

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
