# Peer review of "Climate Change, Agriculture, and Biodiversity: How Does Shifting Agriculture Affect Habitat Availability?"

_land, doi:10.3390/land11081257_

Round 1
Reviewer 1 Report
The manuscript titled “ Climate Change, Agriculture, and Biodiversity: How does shifting agriculture affect habitat availability?“. I find the idea interesting and in line with the aim of the journal. I have some concerns about the experimental setup to justify what the authors claim. Moreover, the rationale behind some of the data presented was not entirely clear. I also recommend to the authors improve their references by conducting a more extensive review of international literature. Particularly, the introduction statements are not supported by the references selected by the authors. The logic of some sentences is also questionable. Below is my point-to-point analysis of the manuscript.
The abstract is not properly written, it should be crisp, and it should contain an introduction aim hypothesis aim result, and conclusion. The introduction section is too long in the abstract; one line of the background of the study in the abstract attracts the reader the most. A connective link is missing between different sections. Also, the concluding part of the introduction is missing at the end of the introduction. The author should make the introduction section crisp and to the point related to research, which I don't find in the present form of the manuscript. The introduction section is the only thing that is weak. I suggest the author to modify the introduction section by a more extensive review of international literature.
Statement „While the human impacts of geographic changes in corn production have received close attention, fewer studies have examined the impacts of a northward-shifting Corn Belt on non-human ecosystem values and services, such as biological diversity. Biodiversity in remnant grassland and shrubland ecosystems in the region, in particular, is globally distinctive“ is a strong statment so it need to be justify by few more references.
Rest is ok especially abstract and introduction section needs improvement. Just an suggestion. I congratulate author for grate piece of work.
Author Response
Thank you for a thoughtful and attentive review. Please see italicized responses below.
The abstract is not properly written, it should be crisp, and it should contain an introduction aim hypothesis aim result, and conclusion. [The abstract has been revised and tightened, and the central aim is more clearly stated now.] The introduction section is too long in the abstract; one line of the background of the study in the abstract attracts the reader the most. [This has been tightened, although two short ideas were needed that required two sentences, rather than one.] A connective link is missing between different sections. Also, the concluding part of the introduction is missing at the end of the introduction. [Brief additional comments have been added.] The author should make the introduction section crisp and to the point related to research, have tried to sharpen and simplify as well as to support statements more - which I don't find in the present form of the manuscript. The introduction section is the only thing that is weak. [Thank you for helpful comments on this.] I suggest the author to modify the introduction section by a more extensive review of international literature. [This was a helpful suggestion, and additional comments and references have been added.]
Statement „While the human impacts of geographic changes in corn production have received close attention, fewer studies have examined the impacts of a northward-shifting Corn Belt on non-human ecosystem values and services, such as biological diversity. Biodiversity in remnant grassland and shrubland ecosystems in the region, in particular, is globally distinctive“ is a strong statment so it need to be justify by few more references. [This is a helpful point; the sentence has been revised to be more focused, and references have been added.]
Rest is ok especially abstract and introduction section needs improvement. Just an suggestion. I congratulate author for great piece of work. [Again thank you for the time and attention in reviewing and improving the paper.]
Reviewer 2 Report
The paper is focused on understanding how agricultural changes due to climate changes affects habitat availability in the Corn Belt region. In my opinion, some adjustments need to be done for improving the quality of the paper:
1. Methodology used should be illustrated more in depth, especially concerning the reasons at the basis of the use of the SDM.
2. Policy implications are illustrated, but it can be useful specifying which policies measures can be pratically adopted. In other terms, which measures can be promoted for achieving the objectives proposed by the authors?
Author Response
Thank you for taking the time to review this paper. These comments were helpful in improving the paper. Please see responses in italics below.
The paper is focused on understanding how agricultural changes due to climate changes affects habitat availability in the Corn Belt region. In my opinion, some adjustments need to be done for improving the quality of the paper:
1. Methodology used should be illustrated more in depth, especially concerning the reasons at the basis of the use of the SDM. – This is a helpful point; the discussion of SDM methods and justifications has been expanded, with additional references.
2. Policy implications are illustrated, but it can be useful specifying which policies measures can be practically adopted. In other terms, which measures can be promoted for achieving the objectives proposed by the authors? – This point has been expanded, with additional comments and references.
Thank you for your time and attention in this review.
Reviewer 3 Report
An interesting paper, but the English language is quite poor, with numerous grammatical errors. For example,
Line 11 - the impacts
Line 15 - the degree
Line 18 - remain
Line 28 - dominates
Line 33 - meat???
Line 50 - why is Leiserowitz et al. underlined?
Line 77 - remove Grassland
Line 84 - what do you mean by 'fell'? How does a program fall?
Line 91 - is shifting
Line 91 - the impacts
Line 97 - the geographic
Line 118 - the distribution
Line 155 - close parentheses
Line 215 - the percentage - twice
Line 217 - the percentage
Line 228 - BIO10
Line 233 - the current
Line 235 - the Canadian
Line 259 - The proportion
Line 260 - the greatest
Line 287 - the abundance
Line 304 - the climate
Line 337 - both biodiversity? both biodiversity and ... ?
Line 343 - is important
Line 346-347 - incomplete sentence
Line 348 - no greater extent? what do you mean?
Line 350 - on cultivation (not in)
Line 353-355 - incomplete sentence
Line 357 - the Canadian
Line 367 - ag? agricultural?
Line 377 - the Farm
Line 396 - shape create?
Line 403 - protecting
Line 409 - this land cover class
Line 411 - the coming
Line 414 - uncultivated ... ?
Author Response
Thank you for a very helpful review, which has greatly improved the paper and has prompted elaboration on some important points. The attention to wording and clarity is greatly appreciated. Please see comments in italics below.
Line 11 - the impacts --corrected
Line 15 - the degree --corrected
Line 18 - remain --corrected
Line 28 - dominates --corrected
Line 33 - meat??? --further explanation has been added, with references
Line 50 - why is Leiserowitz et al. underlined? --corrected
Line 77 - remove Grassland --this sentence was modified
Line 84 - what do you mean by 'fell'? How does a program fall? By at least 20% --further explanation has been added, with references
Line 91 - is shifting --corrected
Line 91 - the impacts --corrected
Line 97 - the geographic --corrected
Line 118 - the distribution --corrected
Line 155 - close parentheses --corrected
Line 215 - the percentage - twice --corrected, thank you for noticing
Line 217 - the percentage --corrected
Line 228 - BIO10 --corrected
Line 233 - the current --corrected
Line 235 - the Canadian --corrected
Line 259 - The proportion --corrected
Line 260 - the greatest --corrected
Line 287 - the abundance --in this case it seemed the previous form was suitable, but if the reviewer feels strongly, this can be changed
Line 304 - the climate --corrected
Line 337 - both biodiversity? both biodiversity and ... ? --corrected
Line 343 - is important --corrected
Line 346-347 - incomplete sentence --corrected
Line 348 - no greater extent? what do you mean? --good point; further explanation was provided
Line 350 - on cultivation (not in) --corrected
Line 353-355 - incomplete sentence --corrected
Line 357 - the Canadian --corrected
Line 367 - ag? agricultural? --corrected
Line 377 - the Farm --in this case, it seemed suitable to keep the adjective form
Line 396 - shape create? --corrected
Line 403 - protecting --corrected
Line 409 - this land cover class --corrected
Line 411 - the coming --corrected
Line 414 - uncultivated ... ? --corrected
Reviewer 4 Report
The Corn Belt is a well-known area, the way this area evolves under the conditions of the predicted climate changes is very important. In this context, I consider that the paper is important, it is well documented.
Also, the way in which the sustainable development of the area is supported by society, through community programs, such as the Common Agricultural Policy at the European level, is also an important aspect that the paper addresses as important in the US conditions.
For me, the paper is clear, but I recommend some minor additions to improve the paper:
- I think that the quality of figure 3 can be improved.
- I recommend that the discussions in the paper be improved with references, in order to support the debated statements.
- I recommend improving the conclusions by summarizing the main ideas.
Author Response
Thank you for your thoughtful attention to this paper. Please see responses in italics below.
- I think that the quality of figure 3 can be improved. --I was not sure what improvements would be helpful in particular, but the figure's size and resolution have been improved.
- I recommend that the discussions in the paper be improved with references, in order to support the debated statements. --the discussion has been revised for clarity, and additional comments with supporting references have been added.
- I recommend improving the conclusions by summarizing the main ideas. -- Thank you, this has been done